# The deletion of *HK-1* gene affects the bacterial virulence of *Pseudomonas stutzeri* LH-42

**Si Shan**[1], **Tingting Hu**[2], **Yu Yang**[1,3]*

**1** School of Minerals Processing and Bioengineering, Central South University, Changsha, Hunan, China, **2** The First People's Hospital of Jingzhou, Jingzhou, Hubei, China, **3** Key Laboratory of Biohydrometallurgy of Ministry of Education, School of Minerals Processing and Bioengineering, Central South University, Changsha, Hunan, China

* csuyangyu@csu.edu.cn

**Data Availability Statement:** All relevant data are within the manuscript and its Supporting Information files.

**Funding:** unfunded studies.

## Abstract

Two-component systems (TCSs) are widespread regulatory systems in bacteria, which control cellular functions and play an important role in sensing various external stimuli and regulating gene expression in response to environmental changes. Among the nineteen genes for the two-component system found in the whole genome of *Pseudomonas stutzeri* LH-42, one of the TCS coded by the *HK-1* gene, has a structural domain similar to the HAMP domain, which plays an important role in regulating bacterial virulence in other bacteria. In this study, the deletion mutant LH-42△*HK-1* was successfully constructed using the lambda Red recombinase system. Compared with the wild-type strain, the mutant strain LH-42△*HK-1* showed a significantly slower growth time and a longer stationary phase time. In addition, in the plate bacteriostatic experiment with *Escherichia coli* DH5α as an indicator strain, the inhibition zone size of the mutant strain showed significantly less than the wild-type strain(P<0.05), indicating that the virulence of the mutant strain was significantly reduced compared with the wild-type strain. Overall, the results indicate that the deletion of the gene *HK-1* decreased bacterial virulence in *Pseudomonas stutzeri* LH-42.

## 1. Introduction

Two-component systems (TCSs) are an important regulatory mechanism in microorganisms, which enable microorganisms to sense environmental changes and respond appropriately to a diverse range of stimuli, including sensing pH, salinity, redox status, osmotic pressure, and antibiotics [1]. A classical two-component system is composed of a signal receptor Histidine Kinase (HK), and Response Regulator protein (RR) (Fig 1) [2]. Normally, HKs consists of three parts: an N-terminal transmembrane sensor domain for signal detection, a DHp domain for potential auto-phosphorylation sites, and a C-terminal CA domain for catalysis and ATP binding. RR usually consists of two domains, mainly the receiver and DNA-binding domains [3]. Following detecting a specific extracellular signal, the HK is auto-phosphorylated at a conserved histidine residue. The phosphate group is then transferred to an aspartate residue on the RR, leading to downstream gene regulation in response to a signal [4]. Two-component systems-related genes are detected to exist in many types of

**Competing interests:** The authors have declared that no competing interests exist.

microbial genomes and play important roles in regulating diverse signaling pathways. For example, GacS/GacA controls bacterial virulence in *Pseudomonas syringae* [5]. QseB/QseC is known to be involved in quorum sensing in *Escherichia coli* [6–10]. However, the mechanisms and function of these Two-component systems are largely unknown in many bacteria.

*Pseudomonas stutzeri* is a number of the genus *Pseudomonas* of *proteobacteria*, characterized as heterotroph known to degrade several hydrocarbons [11]. It has been detected mainly in petroleum-contaminated soil. In this study, *Pseudomonas stutzeri* LH-42 was isolated from petroleum-contaminated soil collected from Liaohe Oil Field, Liaoning, China [12]. The strain *Pseudomonas stutzeri* LH-42 is a gram-negative, chemoheterotrophic bacteria. In previous studies, *Pseudomonas stutzeri* LH-42 has completed the whole genome sequencing and gene function annotation. Through bioinformatics analysis of the whole sequence of *Pseudomonas stutzeri* LH-42, the genome contains genes involved in extensively utilizing carbon source, nitrogen fixation, denitrification, sulfur metabolism, degradation of aromatic compounds, iron acquisition and metabolism, multiple pathways of protection against environmental stress [13,14]. In addition, nineteen genes for the two-component system were found in the whole genome of *Pseudomonas stutzeri* LH-42 (shown in Table 1). There is still, however, many two-component systems functions that were unknown.

Among nineteen genes for the two-component system found in *Pseudomonas stutzeri* LH-42, one of the TCS coded by the *HK-1* gene has a structural domain HAMP (Histidine kinases, Adenylyl Cyclases, Methy Binding proteins, Phosphatases) [15], which is similar to Hamp domains in other bacteria. Fig 2A shown the schematic map of the domain organization of the TCS coded by the *HK-1* gene. The numbers indicate the start and end amino acid positions of the indicated domains. The target protein consists of 452 amino acids, and according to the result of the blast from the NCBI website(https://blast.ncbi.nlm.nih.gov/), the similarity between the TCS coded by the *HK-1* gene, has the highest similarity with other bacterial functional proteins of 44.97% (As shown in Table 2). The amino acid sequences of the target protein were aligned with the amino acid sequences of the five HAMP domains with the highest homology (Fig 2B). In Fig 2B, the target protein is the target protein coded by the *HK-1* gene in this study. 1, 2, 3, 4 and 5 are the amino acid sequences of five HAMP domains with high homology to low homology in Table 2, respectively. And from the figure, we can find that the sequence similarity was high in the middle and back segments. The site marked in Fig 2B (His 241) was identified as a conserved autophosphorylation site by bioinformatics analysis, and the site is located in the Histamine Kinase domain. In addition, combined with literature reports, *Escherichia coli* str.k-12 substr.MG1655 Sensor histidine kinase CusS, which has the highest homology with target histidine kinase, plays a crucial role in bacterial metal detoxification and regulates bacterial virulence [16,17]. Therefore, to identify the relationship between the gene *HK-1* and the bacterial virulence of *Pseudomonas stutzeri* LH-42, understanding the underlying mechanisms is crucial for devising antibacterial treatments. The deletion mutant strain LH-42Δ*HK-1* was constructed and performed bacterial virulence detection.

This study, used lambda Red homologous recombinase system technology to construct the mutant strain [18]. Then, the mutant strain was compared with the wild-type strain to assess growth changes and detect changes in bacterial virulence by Inhibitor zone assay. It's hoped that this study can provide useful information for the following research on the two-component system of *Pseudomonas stutzeri* LH-42. At the same time, it provides some theoretical reference for directional modification of *Pseudomonas stutzeri* LH-42 so that it can still play an effective probiotic effect in different habitats.

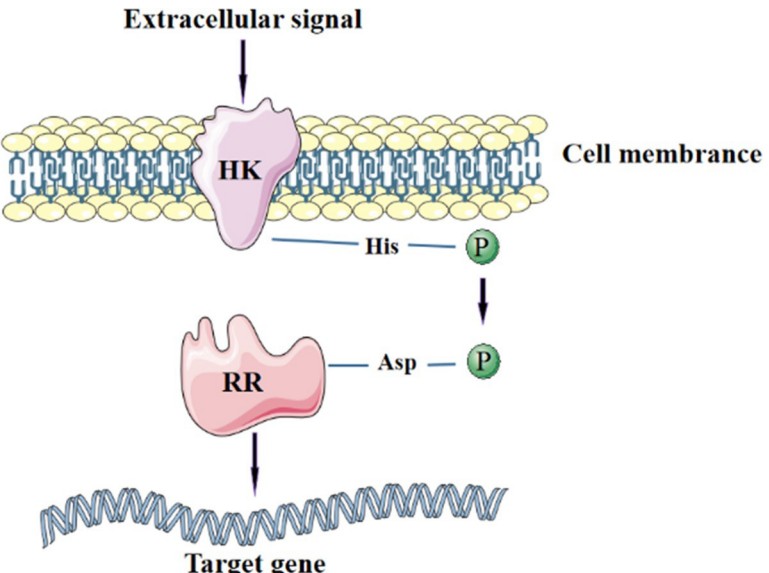

**Fig 1. Mechanism of the two-component system.**

## 2. Material and methods

### 2.1 Bacterial strains, plasmids, media, and growth conditions

In this study, the strain *Pseudomonas stutzeri* LH-42 was cultured in GYP liquid medium with shaking at 30°C and 170 rpm. (GYP liquid medium containing tryptone, 10 g L$^{-1}$; yeast extract,

**Table 1. Two-component systems of *Pseudomonas stutzeri*.**

| Protein name | Hypothesis Gene name | Location |
|---|---|---|
| Two-component Sensor Histidine Kinase | HK-1 | Contig001-15700-17058 |
| Two-component Sensor Histidine Kinase Dcus | HK-2 | Contig016-1882-3489 |
| Sensor Histidine Kinase | HK-3 | Contig50- 15352–16716 |
| Hybrid Sensor Histidine Kinase, partial | HK-4 | Contig55-5637-8274 |
| PAS Domain-containing Sensor Histidine Kinase, partial | HK-5 | Contig129-578-3176 |
| Sensor Histidine Kinase | HK-6 | Contig144-28377-29435 |
| Two-component Sensor Histidine Kinase, partial | HK-7 | Contig160-3087-4222 |
| Two-component Sensor Histidine Kinase | HK-8 | Contig162-4629-5498 |
| Sensor Histidine Kinase | HK-9 | Contig205-784-2052 |
| Sensor Histidine Kinase, partial | HK-10 | Contig207-3290-4451 |
| Two-component Sensor Histidine Kinase | HK-11 | Contig232-1388-2785 |
| Nitrate/Nitrite Two-component System Sensor Histidine Kinase | HK-12 | Contig233-12839-14815 |
| Two-component Sensor Histidine Kinase | HK-13 | Contig235-1677-3278 |
| Two-component Sensor Histidine Kinase | HK-14 | Contig251-5506-6885 |
| Two-component Sensor Histidine Kinase | HK-15 | Contig256-23200-24651 |
| Hybrid Sensor Histidine Kinase | HK-16 | Contig275-10744-12344 |
| Sensor Histidine Kinase | HK-17 | Contig284-6896-8158 |
| Two-component Sensor Histidine Kinase | HK-18 | Contig293-4078-5451 |
| Two-component Sensor Histidine Kinase | HK-19 | Contig295-7351-8724 |

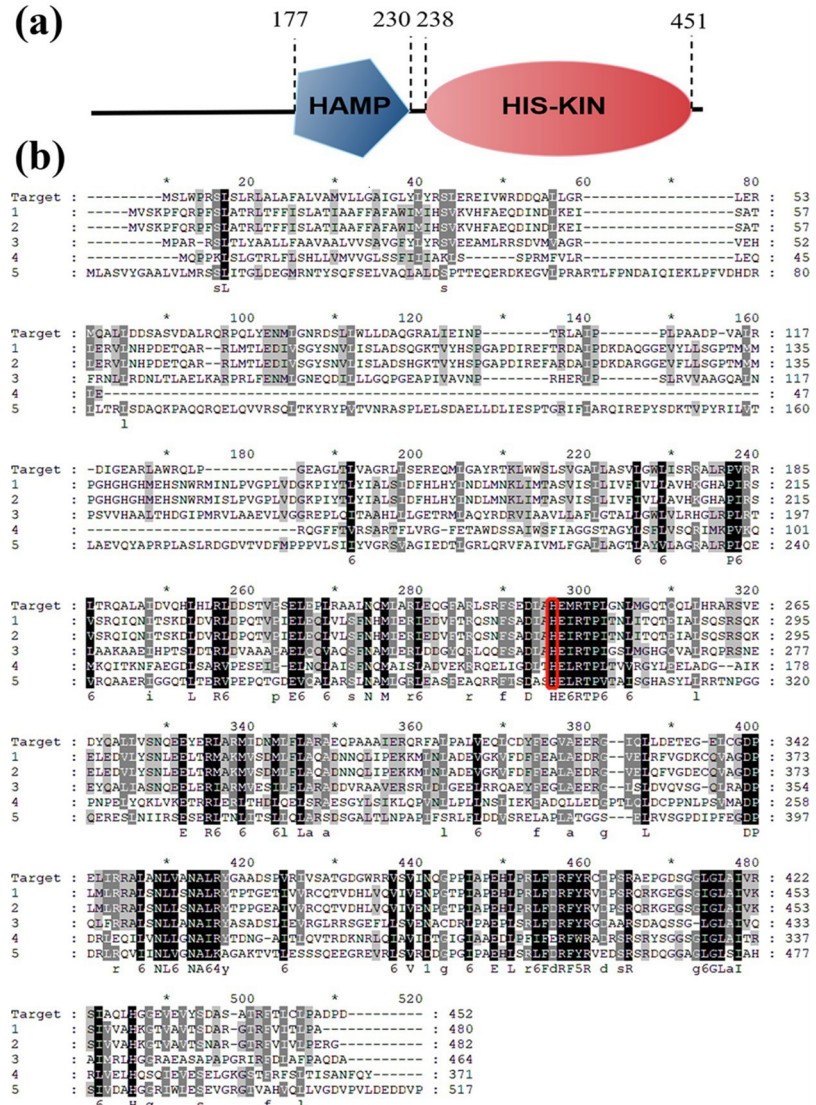

**Fig 2. Domain organization of the TCS coded by the *HK-1* gene.** (a). Schematic map of the domain organization of the TCS coded by the *HK-1* gene; (b). Amino acid sequence alignment of the target protein with five HAMP domains, the target protein is the target protein coded by the HK-1 gene in this study. 1, 2, 3, 4 and 5 are the amino acid sequences of five HAMP domains with high homology to low homology in Table 2, respectively.

5 g L$^{-1}$; glucose, 10 g L$^{-1}$). *Escherichia coli* DH 5α was grown in a Luria-Bertani (LB) medium with shaking at 30°C and 170 rpm. Besides, the plasmids pET-28α and pKD46 used in this

**Table 2. Protein sequence homology alignment results.**

| Homologous species | The degree of homologous (%) |
| --- | --- |
| *Escherichia coli* str.K-12 substr. MG1655 sensor histidine kinase CusS | 44.97 |
| *Escherichia coli* O157:H7 str.Sakai copper-sensing histidine kinase CusR | 44.41 |
| *Pseudomonas aeruginosa* PAO1 two-component sensor | 41.97 |
| *Deinococcus radiodurans* HAMP domain-containing protein | 34.44 |
| *Microcystis aeruginosa* HAMP domain-containing histidine kinase | 32.47 |

study were purchased from Tsingke Biotechnology (Beijing) Co., Ltd. Antibiotics were added at the following concentrations for plasmid selection: Ampicillin 100μg mL$^{-1}$; Kanamycin sulfate 50μg mL$^{-1}$. Bacteria DNA extraction kit (Cat.DP302-02; TIANGEN), Plasmid DNA extraction kit (Cat.DP116; TIANGEN), and gel extraction DNA purification kit (Cat.DP209-02; TIANGEN) were employed.

## 2.2 Construction of deletion mutant strains LH-42△*HK-1*

**2.2.1 Primer design.** This study, used lambda Red homologous recombination system to construct the mutant strains LH-42△*HK-1*. The left and right homologous arms were constructed on both sides of the target gene *HK-1*, and the specific design idea is shown in Fig 3. Primers LF1 and LR1 amplified the left homologous arm, and primers RF1 and RR1 amplified the right. Moreover, the fragment product amplified by the left and right homologous arms was 500bp suitable. Primers LF2 and RR2 are located about 20bp inside LF1 and RR1, which are used for fusion PCR to fuse the left and right homologous arms with resistant fragments. The best length of the target fragment knocked out could be different from that of the inserted resistance gene Kanamycin cassettes (~800bp). In addition, another pair of primers idF and idR were designed on both sides of LF1 and RR1 for sequencing and identification of homologous recombination genes. The primer sequences used to construct the deletion mutant are shown in Table 3.

**2.2.2 Fragment amplified and fusion.** According to the sequence of *P. Stutzeri* LH-42 amplify the left and right homologous arms of the Histidine kinase encoding gene of the two-component system. Then, the plasmid pET-28 as the template to amplify the Km resistant target segments. The PCR mixtures (20μL) contained 1.0μL of template DNA, 0.5μL of Prime-F and Prime-R, 5.5μL of PCR grade water and12.5μL of 2×Taq PCR mix. The primers used in the reaction were LF1, LR1, RF1, and RR1. The PCR cycling conditions consisted of an initial denaturation step at 95°C for 5 min, followed by 35 cycles of 95°C for the 30s, 55–60°C for 60s and 72°C for 2 min and a final extension step at 72°C for 10 min [19]. The primers used in this study are listed in Table 2. And all primers were designed and synthesized by Tsingke Biotechnology (Beijing) Co., Ltd. The PCR products were analyzed by 2% agarose gel electrophoresis. Then, the PCR products were purified and recovered from agarose gels using gel extraction kit. The left

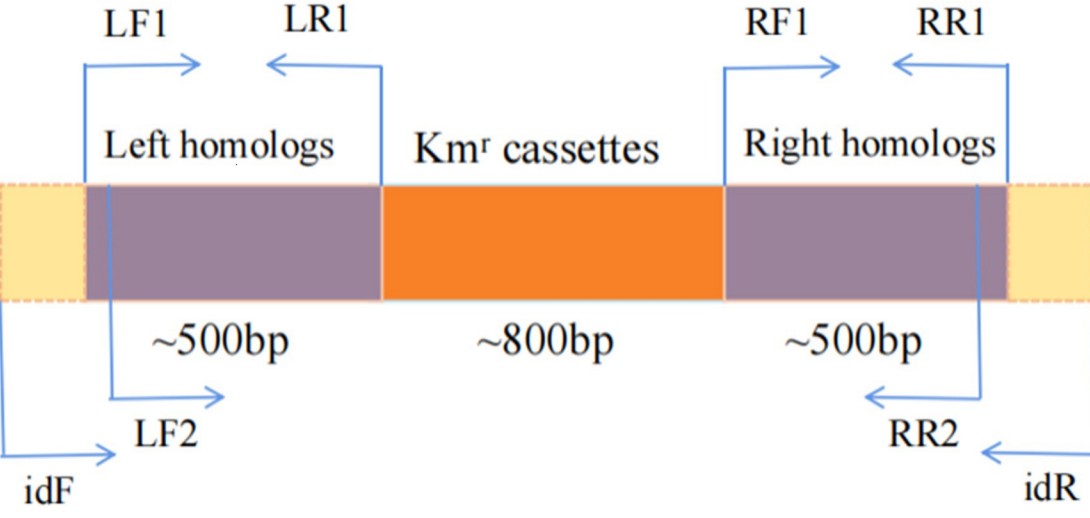

**Fig 3. Primer design.**

**Table 3. Primer sequences.**

| Primer | Sequence (5'→3') |
|---|---|
| LF1 | TGCCAGTGCTGACGGAAATA |
| LR1 | CGCACTGACGTGATGGTTGGCACGGGCAAAAGTCCTG |
| RF1 | ACATGGCATTCGCCTGACTGTGGATGAGCTTGAGCGGGTAC |
| RR1 | GCAAACACCTGATCGAGGAGGAAAC |
| LF2 | GCTGACGGAAATAGTCCCAGAAGG |
| RR2 | CCGACGACTACCTCATCAAACCCT |
| idF | CTGACCACGCACCCAACGATC |
| idF | TGGAGGACGAGGCCAAGACG |

Note: The underlined part of the sequence was used for fusion PCR with Km$^r$ cassettes.

and right homologous arms (460 and 450bp in length) and resistant fragments (~800bp) were fused by PCR to form a homologous fragment, the expected length of the product is about 1800bp. The primers used in the reaction were LF2 and RR2. The PCR amplification system and reaction process are the same as above, and the final DNA concentration was determined.

**2.2.3 Electrical transfer and identification.** The strains *P. Stutzeri* LH-42 were cultured in the log phase, and the competent cells were prepared [20]. The resistant target fragment was transferred into competent cells by the pKD46 electroporation method to construct the mutant strain. The electroporator (BTX ECM399, America) shock mode: 2000V, 12ms, spacing 1mm. Then, heat shock at 42˚C for 10min, and resuscitated at 30˚C for 4h. The mutant strains were screened using resistant media containing Kanamycin and Ampicillin and cultured in the incubator (YiHeng GHP-9050, China) at 30˚C to screen for positive recombinant bacterial colonies. The bacterial colonies growing on the double antibody plates were identified. The single colony was selected and cultured in a GYB medium supplemented with Ampicillin (100μg mL$^{-1}$) and Kanamycin (50μg mL$^{-1}$) until the logarithmic growth stage, and the results were identified using PCR.

## 2.3 Measurement of growth curves

The mutant strain LH-42△*HK-1* and wild-type strain LH-42 were cultured in GYB liquid medium at 30˚C and 170rpm with an initial inoculum size of 2% and initial optical density at 600nm (OD600) of 0.1. Afterwards, a spectrophotometer monitored optimal density at intervals of eight hours, and three parallel experiments were set up for the two groups. Finally, the growth curves of the two groups of bacteria were plotted by Origin.

## 2.4 Inhibitor zone test

The mutant strain LH-42△*HK-1* and the wild-type strain LH-42 were cultured in LB medium at 170rpm at 30˚C, then centrifuged at 8000rpm for 15min at OD600 of 1.2. The supernatant was taken for antibacterial plate test, and *Escherichia coli* DH5α was used as the indicator strain. The *Escherichia coli* DH5α was culture in liquid LB medium to OD600 = 0.6, *Escherichia coli* DH5α were collected by centrifugation at 8000rpm for 10min and then resuspended in 1mL sterile water. 0.1 mL of the bacterial suspension was evenly spread on the solid LB plate with sterilization, 6mm circular filter paper was spread on it, and the supernatant cultures of the mutant and wild strains were dropped onto filter paper. The two groups of experiments were placed symmetrically, and then the plates were placed in a constant temperature incubator at 30˚C for 24h. The size of the bacteriostatic zone was measured with vernier calipers, and the experiment was repeated three times.

## 3 Results

### 3.1 Construction of the LH-42△*HK-1* mutant strain

As shown in Fig 4A, the result of PCR amplification of left and right homology arms, M lane is DNA Marker D2000, Lanes 1–3 were left homologous arm (product length is about 460bp), lanes 4–6 were right homologous arm (product length is about 450bp), clear and bright bands were obtained in lanes 1–6 around 500bp, and there were no other stray bands, the result is the same as expected, indicating that the target bands were obtained. The target band was purified and recycled using a gel extraction DNA purification kit. Agarose gel electrophoresis results of PCR amplified fragment of Km resistance gene are shown in Fig 4B. From gel electrophoresis results, lane M is DNA Marker D2000, and lanes 1–4 were Km resistance genes (product length was about 800bp), clear nucleic acid bands were obtained in lanes 1–4 in the area with a fragment slightly larger than 750bp, and there were no other miscellaneous bands, the result is the same as expected, indicating that the desired target bands were obtained with good primer specificity. As shown in Fig 4C, lane M is DNA Marker D2000, and lanes 1–3 were all homologous fragments, and the product length was nearly 1800bp, the entire fragment comprises an 800bp Km resistance gene, 450bp Right homologous arm and 460bp Left homologous arm. The gene targeting fragment was successfully engineered, the bands length was the same as expected, as shown in lanes 1–3. The target band was purified and recycled. After sequencing, homologous fragments were knocked out by electric transfer.

The colonies capable of growth on Ampicillin and Kanamycin were selected, and the colonies were transferred to a double antibody liquid medium for extended culture. The NCBI comparison and the DNA sequencing results are shown in Fig 5, where the number 1 represents the corresponding sequence of the strains screened by the dual antibody, and the number 2 represents the sequence of the inserted plasmid PET-28 kanamycin resistance gene. The comparison results of the two are completely consistent, which also confirmed the successful construction of the knockout strain LH-42△*HK-1*.

### 3.2 Inactivation of the gene *HK-1* for the two-component system had a certain effect on the growth of strain LH-42

The lambda Red homologous recombinase technique successfully generated the deletion mutant LH-42△*HK-1* was successfully generated by the lambda Red homologous

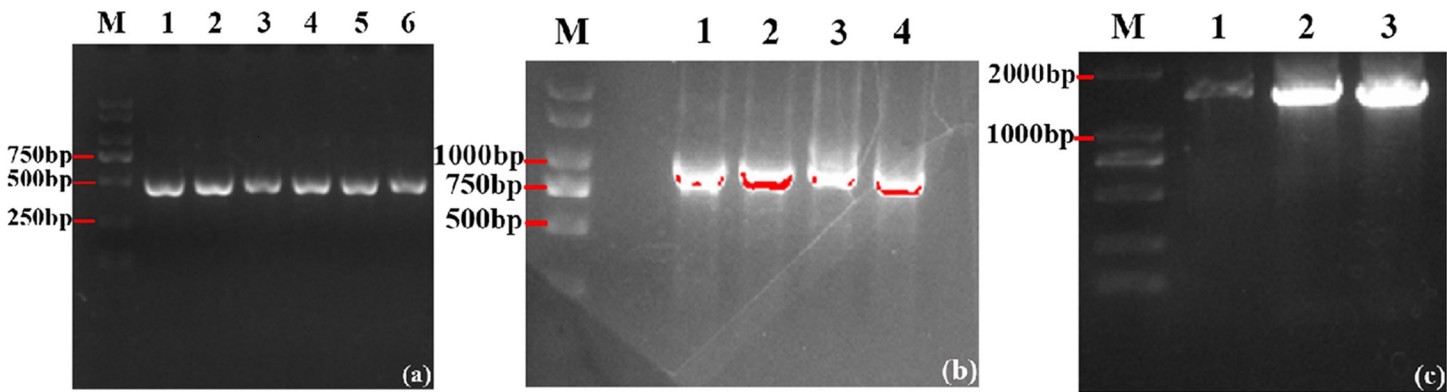

**Fig 4. Results of PCR electrophoresis.** (a) Result of PCR amplification of left and right homology arms; (b) Result of Km resistance gene PCR amplification gel electrophoresis; (c) Results of homologous fragment gel electrophoresis.

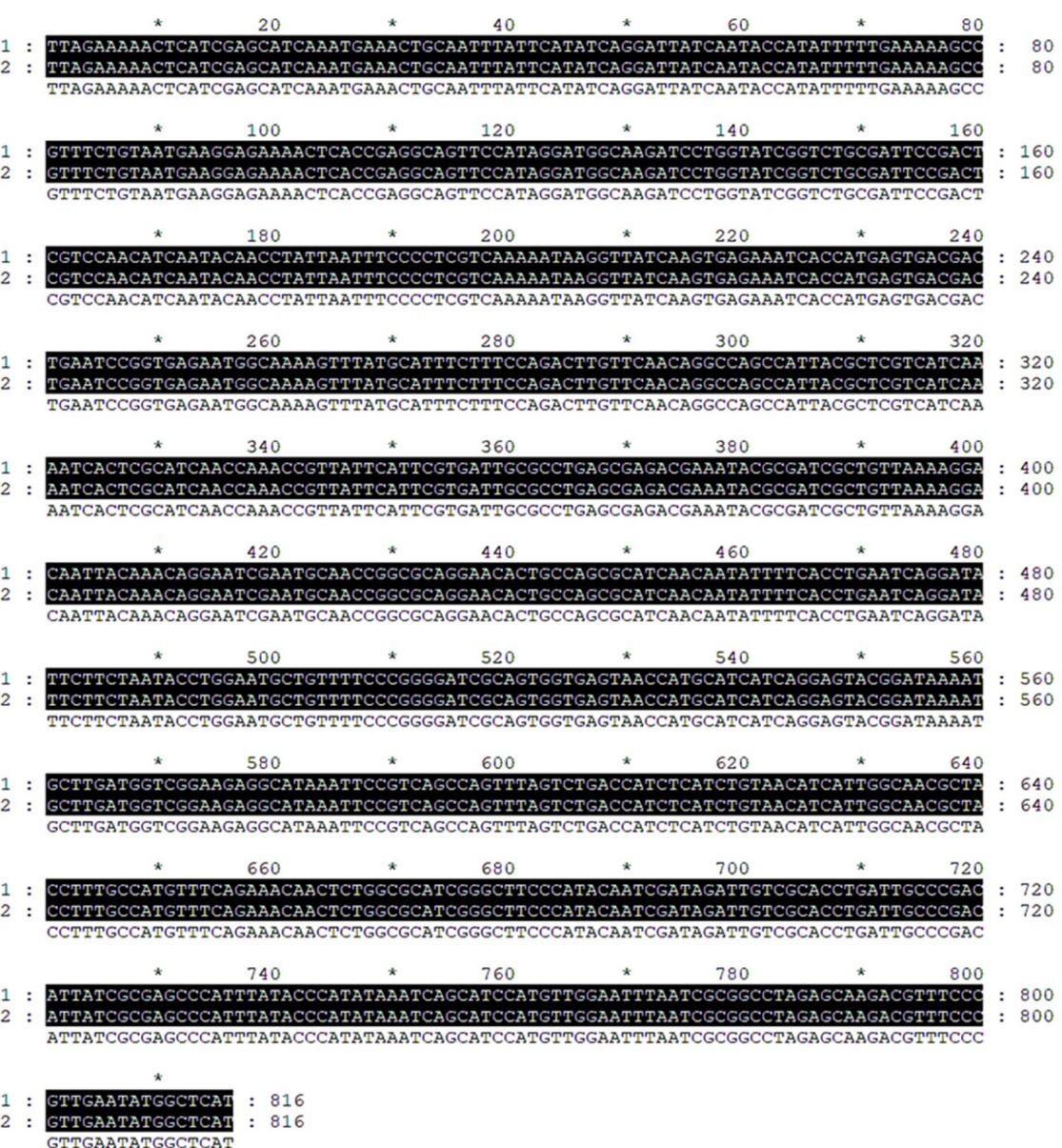

**Fig 5. Sequence alignment of double antibody plate screening strains.**

recombinase technique. Both the wild-type strain LH-42 and the deletion mutant LH-42△HK-1 could grow successfully in the GYP medium. As can be seen from Fig 6, the OD600 of the initial bacterial solution of both wild-type and mutant strains was 0.1. After 40 hours, the maximum concentration of the deletion mutant LH-42△HK-1 was eventually close to that of the wild-type strain. However, the lag phase of the mutant strain LH-42ΔHK-1 was longer than that of the wild-type strain, and the wild strain entered a stationary phase about 32h after culture, while the mutant strain entered about 40h after culture. Therefore, the mutant strain reached the stationary phase about 8 hours later than the wild-type strain.

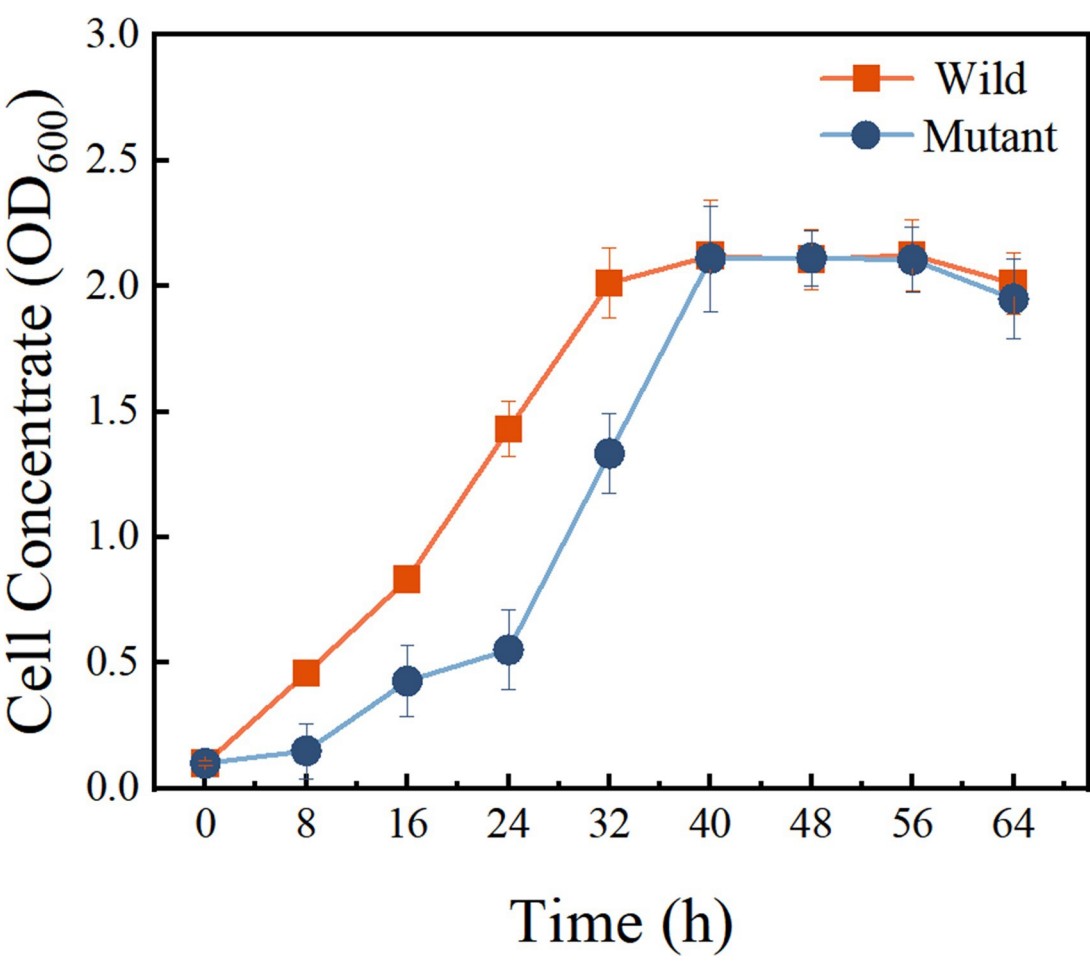

**Fig 6. Growth curve of deletion mutant and wild-type strain.**

### 3.3 Inactivation of the gene *HK-1* for the two-component system influences bacterial virulence

Plate bacteriostatic experiments were carried out on the supernatants of the mutant LH-42Δ*HK-1* and wild strain LH-42. As seen from Fig 7, the supernatant of the mutant strain LH-42Δ*HK-1* and wild strain LH-42 could restrict the growth of indicator strain *Escherichia coli* DH5α, and the size of each inhibition zone was different. The results showed that the inhibition zone size of the mutant strain was significantly less than that of the wild-type strain (As shown in Table 4). According to the t-test of the two groups of data, $p < 0.05$. Therefore, it could be inferred that the gene *HK-1* for the two-component system significantly correlates with bacterial virulence.

## 4. Discussion

Two-component systems are widespread regulatory systems present in bacteria and control the expression of genes mainly involved in secondary metabolism, and they are one of the hotspots in bacterial genomics [21]. The two-component system is important role in regulates some bacterial virulence factors [22]. For example, the *Salmonella enterica* BarA-SirA, the *Erwinia carotovora* ExpS-ExpA, the *Vibrio cholerae* BarA-VarA, and the *Pseudomonas spp*

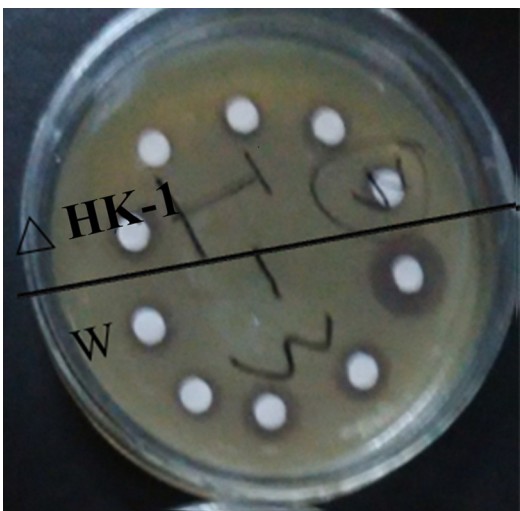

**Fig 7. Results of inhibition zone test.**

GacS-GacA all has been demonstrated that disruption of these two-component system results in a significant reduction in virulence of the bacteria [23–26].

*Pseudomonas stutzeri* is a non-fluorescent denitrifying bacterium widely distributed in the environment [27]. Studies have shown that *Pseudomonas stutzeri* plays an important role in nitrogen fixation, denitrification, organic sulfur degradation, and degradation of aromatic compounds [28–31]. In the early stage of the experiment, the research team analyzed the whole genome sequence of *Pseudomonas stutzeri* LH-42 and found nineteen genes for the two-component system from the whole genome of *Pseudomonas stutzeri* LH-42. One of the genes related to the Two-component system called *HK-1* was selected for study. Bioinformatics analysis showed that the histidine kinase encoded by *HK-1* gene had 44.97% amino acid sequence identity with CusS of the two-component system in *E. coli*. In addition, we identified a DNA-binding response regulator in the gene downstream of the target histidine kinase. The function of the response regulator is to regulate the copper effux system in the two-component system CusS. CusS is a canonical transmembrane histidine kinase [32], and it has been shown that the CusS gene is required for Cu(I)/Ag(I) resistance in *E. coli* [33]. Therefore, we hypothesized that this Two-component related-gene also plays a crucial regulatory role in bacterial metal detoxification and regulation of bacterial virulence.

In this study, the deletion mutant strain LH-42△*HK-1* was constructed and evaluated its growing ability. As a result of the gene knockout experiment, the growth of bacterial strain is affected and the lag phase of bacteria is prolonged. Research by Anne Drumond Villela et al. has also shown that gene deletion delayed growth during the log phase [34]. Besides the significant difference between wild strain and deletion mutant LH-42Δ*HK-1* on the growth during the log phase, the bacterial virulence of the deletion mutant strain and the wild-type strain also exhibited differently. It is not hard to see from the experimental results that the inhibition zone of deletion mutant LH-42Δ*HK-1* was significantly smaller than that of the wild strain.

**Table 4. Statistical results of inhibition zone size measurement.**

| Group | Size of inhibition zone(cm) | | | | | P value |
|---|---|---|---|---|---|---|
| Wild strains(W) | 1.6 | 1.3 | 1.3 | 1.1 | 1.0 | 0.0343 |
| Mutant strains (Δ*HK-1*) | 1.2 | 0.7 | 0.8 | 0.9 | 1.0 | |

However, it still has a certain toxin-producing ability. The results of this study are consistent with those of other literature. In addition, some studies have shown that the knockout of some two-component genes will affect the synthesis of secondary metabolites in bacteria and then affect the virulence of bacteria [35]. Other studies have suggested that bacterial virulence may be affected by the two-component system and have other regulatory mechanisms. The research by Valente et al. on the regulation mechanism of *Pectobacterium wasabiae* virulence showed that in addition to the ExpS/ExpA two-component system, n-acyl-serine lactone's quorum sensing system controlled the virulence of *Pectobacterium* [17]. Therefore, we cannot exclude that other factors under the control of the two-component system contribute to the changes in bacterial virulence in *Pseudomonas stutzeri* LH-42. All of this suggested that in *Pseudomonas stutzeri* LH-42, the gene *HK-1* for the two-component system had a certain effect on bacterial virulence. Furthermore, the lack of the gene for the two-component system delayed the growth of bacteria.

The results of this study will provide a theoretical basis for further understanding of the two-component regulatory system in *Pseudomonas stutzeri* LH-42 and the mechanism of bacterial virulence and also lay a foundation for further research on downstream genes regulated by the two-component system.

## Supporting information

**S1 Raw image. Original images for Blots and Gels.**
(PDF)

## Acknowledgments

Thanks to Rutao Lin for his valuable advice in writing this paper.

## Author Contributions

**Conceptualization:** Yu Yang.

**Data curation:** Si Shan.

**Formal analysis:** Si Shan, Tingting Hu.

**Investigation:** Si Shan.

**Methodology:** Si Shan.

**Resources:** Yu Yang.

**Software:** Tingting Hu.

**Supervision:** Yu Yang.

**Visualization:** Si Shan, Tingting Hu.

**Writing – original draft:** Si Shan.

**Writing – review & editing:** Si Shan.

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
