## [Decision Letter · Decision Letter 0]

2 Sep 2022

PONE-D-22-18322The Deletion of HK-1 gene affects the bacterial virulence of Pseudomonas stutzeri LH-42PLOS ONE

Dear Dr. Yang,

Thank you for submitting your manuscript to PLOS ONE. After careful consideration, we feel that it has merit but does not fully meet PLOS ONE’s publication criteria as it currently stands. Therefore, we invite you to submit a revised version of the manuscript that addresses the points raised during the review process.

The two reviewers, myself and an independent reviewer have commented on the manuscript. Both are requesting a major revision that includes several pieces of added data and information in the text and figures. If you can address these and improve the writing satisfactorily, the manuscript can be re-reviewed.

We look forward to receiving your revised manuscript.

Kind regards,

Michael R Volkert, Ph.D.

Academic Editor

PLOS ONE

Journal Requirements:

"NO authors have competing interests"

5. Please amend the manuscript submission data (via Edit Submission) to include author Tingting Hua.

Additional Editor Comments:

I and an independent reviewer have commented on this manuscript. I agree with the comments of the first reviewer and these should be addressed. In many places it is difficult to understand what is being stated. In addition to the corrections listed in the two reviews, I have one additional suggestion. The extent of homology between the various Hamp domains is stated as a percentage. It would also be useful to include a figure showing the alignment of these homologous domains.

Reviewers' comments:

Reviewer's Responses to Questions

**Comments to the Author**

1. Is the manuscript technically sound, and do the data support the conclusions?

Reviewer #1: Yes

Reviewer #2: Partly

2. Has the statistical analysis been performed appropriately and rigorously? 

Reviewer #1: No

Reviewer #2: No

3. Have the authors made all data underlying the findings in their manuscript fully available?

Reviewer #1: Yes

Reviewer #2: Yes

4. Is the manuscript presented in an intelligible fashion and written in standard English?

Reviewer #1: Yes

Reviewer #2: No

5. Review Comments to the Author

Reviewer #1: The manuscript entitled "The Deletion of HK-1 gene affects the bacterial virulence of Pseudomonas stutzeri LH-42" analyzed the role of histidine kinase HK-1 in the virulence of Pseudomonas stutzeri LH-42. Overall, the manuscript is well written and provided evidence that HK-1 is involved in the pathogenicity of Pseudomonas stutzeri LH-42. I have the following remarks to improve the manscript.

In the introduction, the authors should introduce in general about two component system. These reference documented extensive studies on two component systems (PMID: 32114363, PMID: 31981905, PMID: 31539852, PMID: 35688098, PMID: 35461032, PMID: 34673373, PMID: 32688186, PMID: 32622286, PMID: 31981905, PMID: 31865097

PMID: 35247798, PMID: 34597823, PMID: 33582609 ) and should be cited to promote readership.

A sequence blast of HK-1 with other histidine kinase should be provided so that the audience understand what could HK-1 do in other pathogens.

Statistical analysis should be performed.

The authors should also discuss what could be the response regulator for HK-1.

The authors should also discuss what is virulence traits described in the results section.

Reviewer #2: I am both the editor and am serving as a reviewer, due to difficulty recruiting additional referees. I have not read the other reviewer’s comments and did not do so until my review was completed.

The authors describe the identification of a new virulence gene for Pseudomonas stutzeri by sequence homology, produce a deletion of the HK-1 gene and demonstrate that it produces a smaller zone of inhibition when plated on a lawn of E. coli.

The manuscript requires editing by someone with more experience with English. There are many places where it is difficult to understand the point the authors are trying to make.

The identification and demonstration that the kanR insertion is in the correct location is unclear. For each PCR reaction, it would be useful to state which primers were used, what the expected size should be, and how this compares to the band obtained.

I don’t see the point to the PCR amplification of the homology regions by themselves, as this just shows that this sequence remains in the genome. Since the homology region is in the genome, It would be there even if no insertion was present. This information would be useful only if I am misunderstanding this experiment and one of the primers used to generate these fragments was outside the homology-kanR-homology cassette.

If I understand this section, the only real useful PCR product was the one using primers idF and idR. Sequencing the product could easily confirm the correct structure.

Fig. 4D can be eliminated and simply state in the text that colonies capable of growth on amp and kan were selected for further analysis.

The growth curves text doesn’t state the inoculum size of the two cultures, was it the same:?

The authors also claim that the stagnation is longer. I assume they mean the lag phase. However this is not shown as part of the growth curves, nor is the initial cell density shown as the growth curves begin at 8 hours after inoculation.

The zone of inhibition plate is clear. However, the description of the results is not. It is not stated what concentration of cells is present on each disk. Were different concentrations used on each disk? How many times was this repeated? If several different concentrations were used, then statistics on zone of inhibition needs to be added. The statement that the t-test data was p<0.05 is not helpful without explanation of which set of disks this refers to.

The plate in the picture is labeled deltaTCS and W. The term deltaTCS is not used

6. PLOS authors have the option to publish the peer review history of their article (what does this mean?). If published, this will include your full peer review and any attached files.

Reviewer #1: No

Reviewer #2: No

---

## [Author Response · Author response to Decision Letter 0]

12 Oct 2022

Professor Michael R Volkert and another independent reviewer

Dear Professor Michael R Volkert,

We are submitting a revised manuscript on the web titled “The Deletion of HK-1 gene affects the bacterial virulence of Pseudomonas stutzeri LH-42” (PONE-D-22-18322, EMID: f05411f573cd1896).

Thank you very much for your careful review and the reviewer’s constructive suggestions related to our manuscript. We greatly appreciate the time and positive comments from the reviewers. Those comments are all valuable and very helpful for revising and improving our paper, as well as the great significance to our research.

We have revised the manuscript strictly according to your kind advice and the reviewer’s detailed suggestions. The main corrections in this paper and the responses to the reviewer’s comments are as follows. In the revised submission, the changes to the manuscript are highlighted in red so that they can be easily identified. We appreciate your and the reviewers’ warm work earnestly and hope that the corrections will meet with approval. Please feel free to contact us with any questions, and we are looking forward to your consideration. Thank you very much for all your help.

Kind regards,

Yours sincerely,

Yu Yang

Corresponding authors,

Name: Yu Yang

Key Laboratory of Biohydrometallurgy, Ministry of Education, Central South University

E-mail address: csuyangyu@csu.edu.cn

Responses to Journal Requirements:

1.Please ensure that your manuscript meets PLOS ONE’s style requirements, including those for file naming.

Response/Action: Thank you for the advice. We have made changes based on the PLOS ONE style templates, including raw images, supporting information, manuscript, etc. Thank you for your valuable advice to make the article look more standardized.

2.Please complete your Competing Interests on the online submission form to state any Competing Interests.

Response/Action: Thank you for the advice. I have put the information “The authors have declared that no competing interests exist.” included in my cover letter.

3.PLOS ONE now requires that authors provide the original uncropped and unadjusted images underlying all blot or gel results reported in a submission’s figures or Supporting Information files. 

Response/Action: Thank you for the advice. The blotting or gel image data in this paper have been provided in the supporting information in accordance with the policy requirements.

4.PLOS requires an ORCID ID for the corresponding author in Editorial Manager on papers submitted after December 6th, 2016. 

Response/Action: Thanks for your suggestion. My ORCID ID (0000-0002-2112-7014) is available and has been provided.

5.Please amend the manuscript submission data (via Edit Submission) to include author Tingting Hu.

Response/Action: Thanks for your suggestion. I have added Tingting Hu as the author in the manuscript.

6.Please include captions for your Supporting Information files at the end of your manuscript, and update any in-text citations to match accordingly. 

Response/Action: Thanks for your suggestion. I have referred to the Supporting information Guide to add the document at the end of the manuscript.

Additional Editor Comments to the Author

I and an independent reviewer who has commented on this manuscript. I agree with the comments of the first reviewer, and these should be addressed. In many places, it is difficult to understand what is being stated. In addition to the corrections listed in the two reviews, I have one additional suggestion. The extent of homology between the various Hamp domains is stated as a percentage. It would also be useful to include a figure showing the alignment of these homologous domains.

Response/Action: Thanks for these valuable comments and suggestions. We have revised the manuscript according to your specific comments and added the percentage table of homology degree between different Hamp domains and the comparison map of amino acid sequences of different Hamp domains.

1.Is the manuscript technically sound, and do the data support the conclusions? (Reviewer 1: Yes Reviewer 2: Partly)

Response/Action: Thank you very much for the comments of the two reviewers. Strict control groups and duplication were set in the process of this experiment. Although some data were not uploaded completely before, all the data needed in the experiment have been supplemented in the revised manuscript, and some data are also included in the supplementary information. In addition, thank you very much for your valuable advice.

2.Has the statistical analysis been performed appropriately and rigorously? (Reviewer 1: No Reviewer 2: No)

Response/Action: Thanks for these valuable comments and suggestions. We have revised the manuscript and added the data statistics form and analysis. For example, we added the results of protein sequence homology alignment in Table 2. In addition, part of the data in Table 4 is supplemented. Thank you for your suggestion that help us to improve the quality of our paper.

3.Have the authors made all data underlying the findings in their manuscript fully available? (Reviewer 1: Yes Reviewer 2: Yes)

Response/Action: Thanks for these valuable comments. Uploading all the relevant data of the experiment is what we should do. We also hope that this paper will promote the study of the function of the two-component system of Pseudomonas stuteri LH-42.

4.Is the manuscript presented in an intelligible fashion and written in standard English? (Reviewer 1: Yes Reviewer 2: No)

Response/Action: Thanks for your constructive suggestion. The language problems in this manuscript have been checked carefully. Some language mistakes have been corrected, and also English has been polished. The details of the revised portion were presented in the revised manuscript. Thank you for your keen observation that helps us to find these deficiencies. 

5.Review Comments to the Author

Reviewer 1:

The manuscript entitled “The Deletion of HK-1 gene affects the bacterial virulence of Pseudomonas stutzeri LH-42” analyzed the role of histidine kinase HK-1 in the virulence of Pseudomonas stutzeri LH-42. Overall, the manuscript is well-written and provided evidence that HK-1 is involved in the pathogenicity of Pseudomonas stutzeri LH-42. I have the following remarks to improve the manuscript.

Response/Action: Thanks for these valuable comments and suggestions. We have revised the manuscript point-by-point based on your specific comments.

(1) In the introduction, the authors should introduce in general about two component system. These references documented extensive studies on two component systems and should be cited to promote readership.

Response/Action: Thanks for your constructive suggestion. A general introduction to the two-component system has been added in the introduction, and these references are important for readers to further understand the regulation of Two-component systems. We have cited these references in section 1. In the revised manuscript, we have added examples of Two-component systems regulating different signaling pathways, such as GacS/GacA involved in controlling bacterial virulence in Pseudomonas syringae and QseB/QseC involved in quorum sensing in E. coli. Thank you for your suggestion that help us to improve the quality of our paper. 

(2)A sequence blast of HK-1 with other histidine kinase should be provided so that the audience understand what HK-1 could do in other pathogens.

Response/Action: Thanks for your suggestion. We have added the sequence blast of HK-1 with other histidine kinases in the manuscript. The details can be seen in Fig 2 of the manuscript. And the figure is shown below.

Figure.2 Domain organization of the TCS coded by the HK-1 gene

(a). Schematic map of the domain organization of the TCS coded by the HK-1 gene. The numbers represent the start and end amino acid positions of the indicated domains；(b). Amino acid sequence alignment of Target protein with various Hamp domains. Target was the Target protein coded by the HK-1 gene of this research. 1, 2, 3, 4 and 5 represent the amino acid sequences of the five HAMP domains in Table 2 with homology from high to low, respectively.

(3) Statistical analysis should be performed.

Response/Action: Thanks for your valuable comments and suggestions. We have revised the manuscript and added the data statistics form and analysis. For example, we added the results of protein sequence homology alignment in Table 2. In addition, part of the data in Table 4 are supplemented. 

(4) The authors should also discuss what could be the response regulator for HK-1.

Response/Action: Thanks for your constructive suggestion. In the revised manuscript, we add the predicted possible response regulators of the two-component system and discuss their corresponding functions in the discussion section. For example, the sentence ‘...we identified a DNA-binding response regulator in the gene downstream of the target histidine kinase. The function of the response regulator is to regulate the copper effux system in the two-component system CusS...’. However, we cannot explain the specific regulatory mechanism in detail in this paper, which is also the focus of our next research. Understanding the mechanism of action is extremely important for exploring the function of the two-component system of Pseudomonas stutzeri LH-42. Finally, thank you for your careful guidance to enrich the content of this article.

(5)The authors should also discuss what is virulence traits described in the results section.

Response/Action: Thanks for your constructive suggestion. In the results section of the revised manuscript, we added the description of virulence traits, and in the discussion section, I also added the possible causes of this phenomenon, such as the sentence‘...some studies have shown that the knockout of some two-component genes will affect the synthesis of secondary metabolites in bacteria and then affect the virulence of bacteria...’. In addition, your valuable advice and guidance are greatly appreciated.

Reviewer 2:

The authors describe the identification of a new virulence gene for Pseudomonas stutzeri by sequence homology, produce a deletion of the HK-1 gene and demonstrate that it produces a smaller zone of inhibition when plated on a lawn of E. coli.

Response/Action: Thanks for these valuable comments and suggestions. We have revised the manuscript point-by-point based on your specific comments.

(1)The manuscript requires editing by someone with more experience with English. There are many places where it is difficult to understand the point the authors are trying to make.

Response/Action: Thanks for your constructive suggestion. The language problems in this manuscript have been checked carefully. Some language mistakes have been corrected, and also English has been polished. The details of the revised portion were presented in the revised manuscript. For example, the sentence ‘...the stagnation period of the mutant strain LH-42△HK-1 was longer than that of the wild-type strain..’ has been revised as the following sentence ‘..., the lag phase of the mutant strain LH-42△HK-1 was longer than that of the wild-type strain...’;the sentence ‘...Primers LF2 and LR2 were located in LF1 and about 20bp inside LR1, which were used for fusion PCR to fuse the left and right homologous arms with resistant fragments...’ has been revised as the following sentence ‘...Primers LF2 and RR2 are located about 20bp inside LF1 and RR1, which are used for fusion PCR to fuse the left and right homologous arms with resistant fragments...’; the sentence ‘...Two-component systems related genes are detected to exist in many types of microbial genomes...’ has been revised as the following sentence ‘...Two-component systems-related genes are detected to exist in many types of microbial genomes...’,and so on. Thank you for your keen observation that helps us to find these deficiencies. 

(2)The identification and demonstration that the kanR insertion is in the correct location is unclear. For each PCR reaction, it would be useful to state which primers were used, what the expected size should be, and how this compares to the band obtained.

Response/Action: Thanks for your constructive suggestion. We have shown the sequence alignment map of kanamycin in section 3.1 of the revised manuscript, which provides some evidence for the correct insertion of kanamycin. In addition, for each PCR reaction in the section of experimental methods, we also added the primer used and the expected size. In the section of results, we also added the comparison between the expected result and the actual band. For example, we added the following sentence in the revised manuscript: ‘...The expected length of the product is about 1800bp. The primers used in the reaction were LF2 and RR2...’; ‘...the bands length was the same as expected...’, and so on.

(3)I don’t see the point to the PCR amplification of the homology regions by themselves, as this just shows that this sequence remains in the genome. Since the homology region is in the genome. It would be there even if no insertion was present. This information would be useful only if I am misunderstanding this experiment and one of the primers used to generate these fragments was outside the homology-kanR-homology cassette.

Response/Action: Thanks for your constructive suggestion. The principle of homologous recombination is to replace the target gene fragment with the designed homologous fragment to achieve gene knockout. When the designed homologous DNA fragment is successfully integrated into the target gene fragment, the two fragments will exchange, which will affect the expression of the target gene, and then affect the corresponding function of the target gene so that part of the function is shielded and further affect the organism. Thus, we can speculate on the biological function of the target gene. Although target genes still exist in cells, their expression and function are affected. I don't know if my explanation can make you understand the meaning of homologous recombination. If you have any other questions, please email us.

(4)If I understand this section, the only real useful PCR product was the one using primers idF and idR. Sequencing the product could easily confirm the correct structure.

Response/Action: Thanks for your constructive suggestion. I am sorry that my English expression level is not enough to make it difficult for you to understand this article. As you said, the most important test is the PCR products amplified with the primers idF and idR. In the revised manuscript, we have provided the sequence comparison diagram of the experimental PCR products and the inserted kanamycin template (Fig. 5). The sequence information of PCR products amplified using the primers idF and idR is also provided in the Supplementary information. All these results show that the gene knockout work has been carried out smoothly. The image added is shown below. In addition, your valuable advice and guidance are greatly appreciated.

Figure.5 Sequence alignment of double antibody plate screening strains. The number 1 represents the corresponding sequence of the strains screened by the dual antibody, and the number 2 represents the sequence of the inserted PET-28 kanamycin resistance gene. The comparison results of the two are completely consistent.

(5)Fig. 4D can be eliminated and simply state in the text that colonies capable of growth on amp and kan were selected for further analysis.

Response/Action: Thanks for your suggestion. Fig 4d has been deleted in the revised manuscript, and a simple explanation is given in the result section. Thank you very much for your guidance, which makes this article more concise. 

(6)The growth curves text doesn’t state the inoculum size of the two cultures, was it the same? The authors also claim that the stagnation is longer. I assume they mean the lag phase. However this is not shown as part of the growth curves, nor is the initial cell density shown as the growth curves begin at 8 hours after inoculation.

Response/Action: Thanks for your suggestion. We apologize that some data are not shown in the growth curves of the deletion mutant and the wild-type strain. In the revised manuscript, we have added the initial cell density of bacteria to the figure, and it can be found from the figure that the inoculum size of the two strains is the same. In addition, it was obvious that the mutant strain entered the log growth phase 8h after inoculation, while the wild-type strain entered the log growth phase very quickly after a short lag phase. The following figure shows the details of the changes made to these diagrams. 

(7) The zone of inhibition plate is clear. However, the description of the results is not. It is not stated what concentration of cells is present on each disk. Were different concentrations used on each disk? How many times was this repeated? If several different concentrations were used, then statistics on zone of inhibition needs to be added. The statement that the t-test data was p<0.05 is not helpful without explanation of which set of disks this refers to.

Response/Action: Thanks for your constructive suggestion. I am very sorry that we did not make this part clear in the results. We have added the cell concentration inoculated on the plate and the number of experimental repetitions and other information to the 2.4 part of the manuscript. For example, Add the sentences “the experiment was repeated three times.” and “The Escherichia coli DH5α was culture in liquid LB medium to OD600=0.6, Escherichia coli DH5α were collected by centrifugation at 8000rpm for 10min, and then resuspended in 1mL sterile water. 0.1 mL of the bacterial suspension was evenly spread on the solid LB plate with sterilization, 6mm circular filter paper was spread on it, and the supernatant cultures of the mutant and wild strains were dropped onto filter paper. The two groups of experiments were placed symmetrically” to the manuscript. In addition, we have corrected this error in the revised manuscript regarding the P value. Finally, we also carefully reviewed the entire manuscript to avoid similar mistakes. Thank you for your keen observation to help us find this deficiency. 

(8)The plate in the picture is labeled deltaTCS and W. The term deltaTCS is not used.

Response/Action: Thank you very much for pointing out this mistake. I have corrected the mark on the picture and checked the full text to avoid this mistake happening again. The following figure shows the details of the changes made to the figure.

---

## [Editor Report · Decision Letter 1]

20 Oct 2022

The Deletion of HK-1 gene affects the bacterial virulence of Pseudomonas stutzeri LH-42

PONE-D-22-18322R1

Dear Dr. Yang,

Thank you for your submission and your thoughtful responses to the critiques and for making the requested and modifications to the text, references, figures, and associated files. 

We’re pleased to inform you that your manuscript has been judged scientifically suitable for publication and will be formally accepted for publication once it meets all outstanding technical requirements.

Kind regards,

Michael R Volkert, Ph.D.

Academic Editor

PLOS ONE
---

## [Editor Report · Acceptance letter]

17 Nov 2022

PONE-D-22-18322R1 

The Deletion of HK-1 gene affects the bacterial virulence of *Pseudomonas stutzeri* LH-42 

Dear Dr. Yang:

I'm pleased to inform you that your manuscript has been deemed suitable for publication in PLOS ONE. Congratulations! Your manuscript is now with our production department. 

Kind regards, 

on behalf of

Prof. Michael R Volkert 

Academic Editor

PLOS ONE